# Increasing Gene Editing Efficiency via CRISPR/Cas9- or Cas12a-Mediated Knock-In in Primary Human T Cells

**DOI:** 10.3390/biomedicines12010119

**Published:** 2024-01-06

**Authors:** Natalia Kruglova, Mikhail Shepelev

**Affiliations:** Center for Precision Genome Editing and Genetic Technologies for Biomedicine, Institute of Gene Biology RAS, 119334 Moscow, Russia; mshepelev@mail.ru

**Keywords:** CRISPR/Cas, genome editing, primary T cells, knock-in, HDR, CAR T, HIV gene therapy

## Abstract

T lymphocytes represent a promising target for genome editing. They are primarily modified to recognize and kill tumor cells or to withstand HIV infection. In most studies, T cell genome editing is performed using the CRISPR/Cas technology. Although this technology is easily programmable and widely accessible, its efficiency of T cell genome editing was initially low. Several crucial improvements were made in the components of the CRISPR/Cas technology and their delivery methods, as well as in the culturing conditions of T cells, before a reasonable editing level suitable for clinical applications was achieved. In this review, we summarize and describe the aforementioned parameters that affect human T cell editing efficiency using the CRISPR/Cas technology, with a special focus on gene knock-in.

## 1. Introduction

Genome editing with programmable nucleases that can precisely correct genetic defects or add new functional properties to cells has the potential to cure many human diseases [1]. It relies on programmable nucleases, most notably the CRISPR/Cas technology. Cas nucleases guided by a short RNA molecule introduce a double-stranded break (DSB) into a specific locus of genomic DNA, which is then repaired using cellular machinery via one of several pathways. A DSB can be corrected via non-homologous end joining (NHEJ) that produces small insertions or deletions around the DSB and leads to the disruption of a target sequence, which is used to knock out genes. Alternatively, in the presence of a homologous template DNA, the repair can proceed via a homology-directed repair (HDR) pathway, resulting in the knock-in of the desired genetic material into the genome [2].

One promising target of genome editing is T lymphocytes, which are an important component of the immune system. Currently, there are two main areas of T cell engineering by means of genome editing: oncoimmunology and HIV gene therapy. In the former, T cells are edited to recognize and efficiently kill tumor cells [3]. In the latter, T cells are modified to acquire resistance against HIV infection [4]. In addition to clinical applications, T cell genome editing is an invaluable tool in basic science. Whole-genome knockout and knock-in screens can provide new knowledge about T cell biology and reveal cellular factors, whose manipulation may increase genome editing efficiency or improve the functional properties of T cell products [5,6,7].

Until recently, all manipulations with the cellular genetic material have been limited to the introduction of a target construct using lenti- or γ-retroviral transduction. Although efficient, this process is associated with the risk of insertional mutagenesis, silencing, and unregulated expression. Programmable site-specific nucleases, most notably the CRISPR/Cas technology, have made it possible to directly switch off genes and overcome the limitations of viral transduction. This is achieved via the seamless introduction of a construct to a specific locus in the genome under the control of an endogenous promoter, which should preserve endogenous regulation and a physiological level of expression.

To streamline the site-specific editing of T cells with the CRISPR/Cas technology, several important parameters have been investigated and optimized. These include components of the CRISPR/Cas machinery, namely, the Cas nuclease, as well as the guide RNA molecule, the DNA donor template, and their delivery mode. To preserve T cell functional properties after editing, culturing conditions have been described, including T cell selection, activation, and expansion. Due to these improvements, clinical trials have been launched with gene edited T cells. Dozens of trials, including T cells with the gene KO, have been underway for several years (https://clinicaltrials.gov/search?term=T%20cell%20%20AND%20knock%20out (accessed on 30 June 2023)), and the first data have been obtained [8,9,10,11]. More recently, trials for T cells with the gene KI have been set up (NCT03970382—[12], NCT04213469—[13,14]). In this manuscript, we review the aforementioned parameters that affect human T cell editing efficiency using the CRISPR/Cas technology with an emphasis on gene knock-in. We included recently published studies in this review and highlighted some aspects of the earlier data, which were not the focus of several previous excellent reviews on this topic [15,16,17].

## 2. T Cell Genome Editing for Therapeutic Purposes

We begin by describing two main areas of T cell genome editing and acknowledging the limitations of the clinically approved method of gene insertion using viral transduction. We then highlight features of two Cas nucleases most commonly used for T cell editing, which can potentially overcome the limitations of gene insertion using viral transduction.

### 2.1. T Cells against Tumors

Most manipulations of T cell genetic material aim to produce highly potent anti-tumor lymphocytes. These manipulations are performed ex vivo to direct T cells against cancer cells via either a modified TCR T receptor or a chimeric antigen receptor (CAR) that is specific to a given tumor-associated antigen. In addition, some genes are knocked out or added to improve the functional characteristics of T cells [18]. The Food and Drug Administration (FDA) has approved six drugs based on CAR T cells for the treatment of tumors of B cell origin that target the B cell surface antigens CD19 and BCMA [19]. Similar successes in the treatment of solid tumors have not yet been achieved [20]. Reasons for this include CAR T cell toxicity due to the recognition of normal tissues expressing the same antigen, tumor heterogeneity, and immunosuppressive microenvironment [21]. Furthermore, the widespread use of CAR T cells is limited by the need to produce these cells from the patient’s autologous material, which can be scarce, of poor quality, or require laborious manipulations [20].

To solve these problems and produce T cells with improved functional characteristics, different alterations to the T cell genome have been suggested [20,22]. New genes can be introduced, such as the genes for the CAR and TCR molecules. Moreover, T cells can be equipped with receptors for chemokines expressed by tumor cells or with enzymes that degrade the intracellular matrix to increase tumor infiltration. It has been reported that T cell functions can be further improved via the introduction of genes that code for various cytokines, such as IL-12, 15, 18, and 7, which help T cells to counteract immunosuppressive microenvironments. If necessary, some constructs, such as tEGFR, tNGFR, or the dihydrofolate reductase L22F/F31S, can be inserted in conjunction with the CAR T or TCR T molecules to allow for the selection of successfully modified cells or their targeted killing in vivo [23]. Currently, viral transduction is the only clinically approved method for the insertion of these constructs [20].

Apart from the addition of new elements to the T cell genome, some alterations require the inhibition of gene products. When transduction was the only available method, indirect blockage with dominant-negative constructs or decoy molecules was used [20]. Nowadays, programmable nucleases can switch off genes directly. For example, PD-1 knockout is widely applied to help CAR T cells to withstand immunosuppressive microenvironment [24]. Recently, the knockout of several different molecules has been suggested for the same purpose, including TGFβR2 [25] and chromatin remodeling factors identified in a CRISPR screen [5]. The knockout of HLA and endogenous TCR T molecules can help to produce universal off-the-shelf CAR T cells and solve the problem of autologous T cell products [22].

In contrast to viral transduction, programmable nucleases permit a simultaneous knockout of a gene and the introduction of a new gene construct via knock-in. With this strategy, the gene of a CAR T or any other required molecule can be introduced into the locus of a gene to be knocked out and expressed by an endogenous promoter. One popular target locus is the *TRAC* gene, whose knockout disables the synthesis of endogenous TCR. Eyquem et al. used the *TRAC* locus for the introduction of the CAR gene [26]. To disrupt endogenous TCR, Roth et al. used a modified TCR against NY-ESO-1 into the *TRAC* locus, the constant region of the *TRCB1/2*, or both loci [27].

### 2.2. T Cells against HIV

HIV gene therapy is the second popular area of T cell genome editing [28]. Currently, patients with HIV are treated with life-long highly active antiretroviral therapy (HAART). HAART has several disadvantages, including high cost, strict medication regime, toxicity, and the development of viral escape mutants. Programmable nucleases can potentially help to achieve functional cure or even complete HIV elimination from the body and thus relieve the patient of the need to take HAART [29]. Two main approaches for the gene therapy of HIV are: (1) the production of CAR T or TCR T cells targeting HIV-infected cells, and (2) the generation of CD4+ T cells that are resistant to HIV infection. Several clinical trials are underway (https://clinicaltrials.gov/search?cond=HIV&term=gene%20therapy, accessed on 30 June 2023).

CAR T or TCR T cells have the potential to reduce the volume of HIV reservoir, either by destroying infected cells with low levels of expression of viral proteins during a latent infection or by killing productively infected cells during stimulated provirus reactivation. The HIV envelope protein is the main target of these cells, and different CAR variants have been developed against it. Whereas some CARs are based on the CD4 molecule [30,31], others either include domains from broadly neutralizing antibodies [32] or represent a combination of domains from both CD4 and a broadly neutralizing antibody [33]. CD4-based CAR makes T cells susceptible to HIV, and hence the special measures discussed below have to be taken to protect the cells [28]. TCR T cells can be directed against intracellular antigens, such as gag (NCT00991224).

In comparison to the production of CAR T or TCR T cells against HIV, more attention is given to modifying CD4+ T cells to make them resistant to infection because they represent the main target of HIV. The first such approach is based on gene knockout. It involves disrupting the HIV coreceptors CCR5 or CXCR4, or both [34,35], or eliminating the HIV provirus from infected CD4+ T cells [36]. The second approach includes the introduction of new constructs into the genome to supply T cells with factors preventing viral entry or replication [37,38]. A recent review provides the details of these strategies [39].

In most studies, the discussed constructs are introduced via transduction; however, all the previous considerations in relation to the use of CAR T cells against tumors are also valid. With programmable nucleases, these constructs can be introduced into a particular locus in the genome with a simultaneous knockout of the gene. There are only a few examples of the introduction of an anti-HIV gene via knock-in [37,40]. Our group inserted a construct coding for a GPI-anchored HIV fusion inhibitor into the *CXCR4* locus or into the HIV provirus [37]. Sather et al. introduced a CD46-coding construct into the *CCR5* locus [40].

### 2.3. Comparison of Viral Transduction and CRISPR/Cas Technology for T Cell Genome Editing

The main and only clinically approved way to introduce a target construct into T cells is transduction with a γ-retro or lentiviral vector. This method has several disadvantages. First, despite the development of increasingly safe reporter vectors, there is a risk of insertional mutagenesis [41]. Fortunately, no cases have been reported of tumor transformation of CAR T cells obtained via transduction and introduced into a patient [42,43]. However, there is evidence that CAR T cell clones with a certain integration locus can gain a selective advantage [44]. Second, the expression of a target construct may be far from optimal despite a search for different promoters. When inserted into a random locus in the genome, a construct does not undergo endogenous regulation, and the physiological level of expression is not maintained. An excessively high level of expression from a strong promoter can disrupt the functional properties of cells or expression may become silenced [45]. Third, due to the random integration site, the cell product is highly heterogeneous and varies from donor to donor [8,46,47]. This makes it difficult to analyze results of clinical trials and predict the effectiveness of therapy for a given patient. Fourth, the size of a construct is limited by the vector’s cargo capacity [42]. Finally, if large numbers of therapeutical constructs have to be tested, the production of viral stocks suitable for clinical applications becomes a daunting task. An illustration of this could be the selection of neoTCR T cells with receptors against patient-specific neoantigens. This process includes the production and assessment of several cell products and ideally allows researchers to choose a mixture of several neoTCR T cell clones [12,48].

In theory, the knock-in via programmable nucleases, particularly the CRISPR/Cas technology, can help to overcome these limitations. In contrast to transduction, the CRISPR/Cas-mediated knock-in allows for the introduction of a target construct at a given locus in the genome, which may preserve endogenous regulation and support a physiological level of expression without silencing. Data supporting these assumptions have been obtained in the work on both CAR T cells against tumors and T cells resistant to HIV [26,37,49,50,51].

The first data were obtained in the study by Eyquem et al., who inserted the CAR T gene into the *TRAC* locus and compared this approach to transduction [26]. This and subsequent papers demonstrated that targeted knock-in resulted in more homogeneous CAR expression on T cells when compared to transduction [26,52]. The level of CAR on the cell surface was lower and physiological regulation was retained, which was important for CAR T cell survival and the slowing of their transition to an exhausted state [26,49]. Some evidence suggests that CD4+ T cells are more amenable to lentiviral transduction than CD8+ T cells, whereas modification via CRISPR/Cas-mediated knock-in leads to equal editing of both CD4+ and CD8+ T cells, resulting in higher CD8+/CD4+ ratios, which is important for tumor control [13]. Some research groups did not observe functional differences between transduced T cells and the T cells with knock-in in in vitro conditions. However, the knock-in T cells in vivo demonstrated more efficient tumor killing and maintained the CAR receptor on the cell surface for longer periods of time [26]. Others reported functional differences both in vitro and in vivo in favor of T cells with knock-in [13,50]. The decreased functional properties of transduced T cells have been partially attributed to high CAR expression and a lack of its endogenous regulation [26].

The importance of endogenous regulation and homogenous expression of a target construct has been demonstrated for not only CAR T and TCR T cells, but also for T cells with acquired resistance to HIV [37,51]. Désaulniers et al. showed that the endogenous HIV restriction factor TRIM5α, with two point mutations introduced via knock-in, was upregulated in response to IFNβ, which provided better protection compared to mutant TRIM5α expressed from a transduced construct [51]. Our group observed that the GPI-anchored HIV fusion inhibitor MT-C34, inserted into the *CXCR4* locus under an endogenous promoter, expressed more homogeneously and at a higher level than a transduced construct, as well as conferring better protection against HIV [37].

An additional advantage of T cell editing using the CRISPR/Cas technology, when compared to transduction, is the combination of knockout and knock-in, where the target construct is introduced into the locus of the gene to be switched off. The CAR receptor is predominantly inserted into the *TRAC* locus [13,23,26,27,49,53,54,55,56]. Other examples include *PDCD1* [13,53], *TRBC* [53], *ILR2A* [54], and *B2M* [13,23]. In the case of T cells resistant to HIV, a protective construct can be inserted into the gene of the HIV coreceptors CXCR4 [37] or CCR5 [40].

One important consideration for the production of cells with genome editing is the percentage of cells with the desired modification that would be enough to achieve a therapeutical benefit. Interestingly, the level of transduction in CAR T or TCR T cell products varies from 2% to 96%, leading to different numbers of modified T cells being infused into a patient [8,46,47]. The level of transduction correlates with the differing number of vector copies per cell (vector copy number, VCN) [57]. The standard approach for measuring VCN only provides an estimate for the entire population and does not reveal cell heterogeneity and the presence of cells with high VCN. Single-cell analysis can help to measure this heterogeneity and detect clones with up to 44 vector copies per cell [58]. An enriched population of edited T cells, desirable for clinical applications, can be achieved via a high level of transduction, but this will inevitably increase the proportion of cells with multiple copies of a genetic construct. As noted above, excessively high expression levels can lead to a decrease in the functional characteristics of cells [26,49,50]. We could not find strict recommendations on the level of transduction (% of cells with the desired modification) for a clinical-grade cell product. The strict regulation of VCN is also lacking. The FDA recommends adhering to a threshold of 5 transgene copies per genome and generally aiming for VCN reduction to prevent potential genotoxicity and oncogenicity (presentation by Dr. Vatsan/FDA at ISBioTech conference on 7 March 2017) [59]. Taken together, viral transduction produces a highly heterogeneous cell product whose properties cannot be fully predicted. In theory, when cells are edited with site-specific nucleases, there is no upper limit to the level of editing, analogous to the percentage of transduced cells associated with a particular VCN. As a result, a more enriched and homogeneous population can be generated.

## 3. Genome Editing with the CRISPR/Cas Technology

Four main groups of programmable nucleases are currently used as genome editing tools: meganucleases, ZFNs, TALENs, and CRISPR/Cas [60]. All these groups have been applied to T cell editing. However, the CRISPR/Cas system has become the most widely used due to its simple programming method. Many CRISPR/Cas systems have been described [61]. The CRISPR/Cas system class 2 type II has been the first one used in mammalian cells and is currently the most popular editing tool. Over the past few years, the CRISPR/Cas system class 2 type V-A has gained extensive attention. In both systems, a single Cas protein is required for the target’s recognition and cleavage [61].

In the CRISPR/Cas9 system, the key nuclease is the Cas9 protein, with a size of approximately 1500 aa. The native Cas9 uses a two-part guide RNA molecule consisting of crRNA and tracrRNA, which can be combined into an artificial sgRNA molecule of about 100 nt [62]. Cas9 uses a PAM ‘NGG’, located on the 3′ end of the protospacer, and cleaves the DNA heteroduplex simultaneously by two catalytic domains at a distance of 3 nt upstream from the PAM with the formation of blunt ends [63].

Later, the Cas12a protein from another CRISPR system was described [64]. It has the size of 1200–1500 aa depending on the bacterium species and recognizes a T-rich PAM (TTTV), located to the 5′ end of the protospacer. Cas12a does not require a tracrRNA and uses only a crRNA of about 40 nt. Cas12a has a single nuclease domain and cleaves two DNA strands sequentially, leaving sticky 5′-ends of 4–8 nt in length. The cut occurs at a distance of about 18 nt from the PAM (nearest gap). It is suggested that such a cleavage pattern may support editing via the HDR pathway: Cas12a will continue rounds of target binding and cutting until either homologous recombination with a repair template disrupting the spacer occurs or significant damage accumulates as a result of NHEJ [65].

Despite the several advantages over viral transduction discussed in the previous section, insertion of desired genetic constructs in the genome of T cells via the CRISPR/Cas-mediated knock-in has its own limitations that should be acknowledged. The most crucial of them are efficiency and specificity. The efficiency of knock-in should be high enough to provide therapeutical benefits for patients. Approaches to increase knock-in efficiency will be reviewed in this manuscript. An important point related to efficiency is the availability of genomic targets in relevant loci, which is constrained by the dependance of all native Cas nucleases on PAM. Engineering of Cas nucleases with relaxed PAM requirements or even “PAMless” recognition will help to solve this problem [66]. The specificity issue is the major concern in the genome editing field in general and tremendous efforts are being made to predict and reduce off-target effects. As this issue lies beyond the scope of our manuscript, we refer the readers to several recent reviews on this topic [67,68].

In summary, we conclude that T cell genome editing constitutes an important area of research with significant clinical implications, most notably with respect to treating tumors and fighting HIV infection. Site-specific nucleases, especially the CRISPR/Cas technology, allow for the knockout of genes that hamper T cell functioning. However, in clinical settings, new constructs such as a CAR receptor are still introduced via viral transduction. The CRISPR/Cas technology can potentially overcome the limitations of transduction. In the next section, we describe various approaches to improving the efficiency of T cell editing using CRISPR/Cas, particularly the efficiency of knock-in via HDR.

## 4. Approaches to Increase Knock-in Editing in Primary T Cells

There have been studies of various factors affecting the editing efficiency of primary human cells, in particular T cells, using CRISPR/Cas-mediated knock-in. These factors can be divided into two main groups: (1) the composition and delivery mode of CRISPR/Cas components, and (2) the state of cells that should be amenable to editing. All factors discussed below are indicated in Table 1.

### 4.1. Choice of CRISPR Components and Method of Their Delivery into Cells

This section focuses on the choice of a guide RNA, Cas nuclease, and a donor DNA construct, as well as methods for their delivery. In most cases, we refer the reader to studies where editing was performed in primary T cells. The most significant papers are summarized in a Appendix A, which contains key information about CRISPR/Cas components, their delivery, and the achieved level of editing.

#### 4.1.1. gRNA

Various parameters of gRNA have been investigated to improve the level of cleavage and the efficiency of knock-in in particular (see Figure 1).

Selection of gRNA

The level of knock-in primarily depends on the efficiency of cleavage of the target locus by the Cas/gRNA complex. Therefore, it is important to choose a good gRNA spacer sequence [69]. Various online software tools are available for gRNA selection [70]. Ideally, several candidates have to be experimentally verified because the efficiency of cleavage depends on many factors, not all of which can be accounted for in silico. Such factors include global [71] and local [72,73] chromatin states, the presence of certain motifs in the protospacer [74], and complementary sequences in the spacer and the scaffold of a gRNA molecule [75]. Nucleotide polymorphisms in the target region can also affect editing [76,77]. One study performed on primary T cells reported that combined data on the epigenetic profiles of T cells and in silico gRNA prediction provided a much more accurate estimate of gRNA potency [78].

In general, the higher the level of editing is at a given locus, the higher the expected rate of knock-in via HDR will be [79]. However, two recent papers analyzing indel profiles in primary T cells showed that the HDR pathway outcompeted the MMEJ pathway, but not the NHEJ pathway [79,80]. Deletions of 3 nt and longer that served as an MMEJ signature predicted the level of knock-in better than the overall level of indels [80]. In the presence of a donor DNA, the percentage of MMEJ signatures decreased, and this percentage reflected the highest possible level of HDR for a particular gRNA [54,79,80].

Another important factor is the distance from the cleavage site to the position of the introduced construct relative to the homology arms. The smaller it is, the higher the knock-in level that can be achieved. The optimal position of the insertion site is within 10 nucleotides of the cut site [81].

2.Editing with multiple gRNAs

To increase the level of editing, several studies have proposed using two or three gRNAs per gene. The gain is not usually substantial compared to the most efficient gRNA alone [82,83,84], although it may be advantageous for epigenetically closed loci [78]. Furthermore, multiple DSBs are known to increase the likelihood of large chromosomal rearrangements, and thus this method of increasing the level of editing is not the most reliable [53,85,86].

A novel “double tap” method has been proposed, in which additional gRNAs target the Cas nuclease to the most common indel variants for a given locus produced by NHEJ [87]. This “secondary” editing permits the conversion of a large proportion of NHEJ events into HDR. Another study subsequently validated this method in different cell types, including primary T cells [88]. “Recursive” editing doubled the level of knock-in of αBCMA CAR into the *TRAC* locus [88]. The effect of additional gRNAs on off-target editing has not been analyzed.

3.gRNA delivery and protection from degradation

gRNA can be delivered to cells as an expression plasmid, an in vitro transcribed (IVT) molecule, or a chemically synthesized molecule. Plasmid DNA has been found to be toxic to primary T cells [69] and thus represents a suboptimal delivery molecule. IVT or chemically synthesized gRNA molecules are degraded in the cell in the absence of the Cas nuclease and therefore must be delivered as a complex with it [89,90]. When Cas9 was delivered to primary T cells in the form of mRNA together with IVT gRNA, the level of editing of the *TRAC* locus was less than 10%; however, when gRNA was re-delivered the next day, the level of editing increased to 90–95% [91]. Another way to protect gRNA from degradation is to use chemically modified gRNAs, as shown in the seminal work by Hendel et al. [90]. Various modifications to gRNA bases have been proposed to increase their stability and block the activation of intracellular RNA receptors, which are described below (details can be found in the review by Allen et al. [92]). Since then, most of the work on primary T cell editing has used modified gRNAs (see Appendix A).

#### 4.1.2. Cas Nuclease

Among various Cas proteins, the SpCas9 nuclease was the first one to be described and tested for editing the mammalian genome [62,93,94]. Therefore, it is not surprising that most studies on primary T cell editing have used this nuclease [12,27,35,49,54,55,95,96]. Recently, the nuclease Cas12a has been applied to edit primary T cells [97,98,99,100,101,102]. The type of a nuclease, its delivery mode and targeted nuclear transport have been investigated to improve knock-in levels (see Figure 2).

The type of nuclease

The need to find the optimal nuclease with high activity and high accuracy stimulates a constant search for new Cas proteins [103]. In addition, the use of different Cas nucleases expands the number of available target sequences due to different PAMs of Cas proteins from different systems. This is especially important when a seamless insertion of a construct into a particular locus is required, which imposes severe restrictions on the choice of target sequences for guide RNAs. In addition to these obvious reasons, it is assumed that, by comparing different nucleases, it will be possible to find a variant that provides a higher HDR/NHEJ ratio and gives a higher level of knock-in.

Since the resection of DNA ends must occur with the formation of free 3′-strands in editing via the HDR pathway [104], it has been suggested that nucleases leaving sticky 3′-ends of DNA can shift the HDR/NHEJ balance during DSB repair in favor of HDR [40]. Indeed, a higher level of knock-in in primary T cells was observed with the MegaTAL nuclease that leaves 3′-overhangs, compared to the TALEN nuclease which leaves 5′-overhangs [40].

Overall, primary T cells have been edited with all available site-specific nucleases, including the homing endonuclease I-CreI [105], MegaTAL [40], TALEN [40], ZFNs [106], Cas9 (Appendix A), and Cas12a (Appendix A). Among these, the homing endonucleases I-CreI [105] and MegaTAL [40] leave 3′-overhangs, TALEN [40] and Cas12a [65] leave 5′-overhangs, and Cas9 produces blunt ends [63]. In all cases, protruding ends may have different lengths. However, according to the results of many research groups that achieved high rates of editing in primary T cells, no relationship is observed between the type of nuclease and the level of knock-in [40,97,103,105,106,107].

Another suggestion was made when comparing the distance from the PAM to the cut site and the resulting indel profiles for Cas9 and Cas12a [108]. Since the Cas12a nuclease makes the nearest DNA break at a distance of about 18 nt from the PAM, it is capable of several rounds of target recognition and cutting, so long as the protospacer is not affected by significant changes. It is assumed that in the presence of a template DNA, such a cutting pattern should lead to the enrichment of cells with a locus edited via the HDR pathway. However, experimental data do not allow us to unequivocally confirm or refute this. In some studies, the level of knock-in with Cas12a was higher than with Cas9 [98,100], while in others it was the same or lower [49,97]. Perhaps, the level of knock-in depends on the choice of specific guide RNAs, the target locus, properties of the isolated proteins (NLS, tags), and the method of their delivery.

2.Type of molecule for delivery

The Cas nuclease can be delivered to cells in various forms: as a plasmid DNA, mRNA, or protein. The editing of primary T cells with Cas9 delivered as a plasmid was ineffective [9,90,109]. However, even with the low level of editing and the technologies available in 2015, the PD-1 knockout T cells obtained using plasmids were tested in clinical trials. The knockout rate was about 20% when detected via Sanger sequencing of clones and about 5% when calculated via NGS. The authors note that, despite the detected off-target effects (at the level of 0.05%), no critical side effects were observed in patients [9].

In general, T cell editing with Cas9 in the form of an mRNA or in the form of a protein in the RNP complex produces similar results [110]. However, the RNP form is used much more frequently for a number of reasons. First, for RNP, in contrast to the mRNA molecule, it is easier to select the required dose, limit the duration of the nuclease activity due to the rapid turnover of the protein in the cell, and, as a result, reduce the risk of off-target effects [69]. Second, the RNP form is much less toxic to cells and less likely to activate the innate response of cells to foreign nucleic acids in the cytoplasm [69,111]. Third, if mRNA is used for Cas9 delivery, the instability of the guide RNA in the absence of the protein must be taken into account, as described in the previous section. If gRNA and the Cas protein are administered as RNP pre-assembled in vitro, then the probability of rapid gRNA degradation is greatly reduced. Fourth, donor DNA must be present in the cell when the nuclease is maximally active, which does not occur if the donor and mRNA for the nuclease are delivered simultaneously. Yang et al. showed that linear dsDNA in the cell is rapidly degraded, with a half-life of about 1 h [111]. The authors electroporated mRNA for TALEN into primary T cells and observed that the level of CAR receptor knock-in increased 15 times if the donor dsDNA was delivered when the TALEN protein level in the cell reached its maximum. To achieve this, the authors added the donor DNA during the second electroporation, which was performed 16–20 h after the first one. With RNP electroporation, this problem did not occur. Finally, it was shown using 293T cells that the activity of the Cas9 protein is blocked by endogenous RNA molecules which compete with gRNA for nuclease binding [112]. Therefore, the most optimal conditions for the activity of Cas9/gRNA (maximum activity normalized to the protein level) are achieved when the complex is formed before entering the cell. Thus, the RNP form appears to be the most preferable.

3.Nuclease delivery into primary T cells

Electroporation has been proved to be the most effective way to deliver genetic constructs and proteins into primary T cells [35]. The majority of studies performing primary T cell editing use electroporation and deliver gRNA and the Cas nuclease in the form of RNP complexes (see Appendix A). The most commonly used device for electroporation is the 4D Nucleofector (Lonza, Basel, Switzerland) with the EH115 program [49,54,55,83,95]. With this device, it is possible to achieve 90–100% knockout of selected loci with a cell survival rate of 60% to 90%, or even higher [52,54,55,98]. Other devices that are used much less frequently include the Neon Transfection system (Invitrogen, Thermo Fisher Scientific, Waltham, MA, USA) [35,97] and the MaxCyte electroporator (Miltenyi Biotec, Bergisch Gladbach, Germany) [23]; both give a high knockout level. Additionally, there are other devices capable of producing equally good results [26,113]. Recently, VanderBurgh et al. developed a flow electroporation apparatus in a very thin flow cell and achieved 90% TRAC knockout on primary T cells with preserved survival [114].

The manufacturers of the 4D Nucleofector device (Lonza, Basel, Switzerland) note that, due to the device’s selected pulse parameters and buffer composition, it can cause not only electroporation, but also nucleofection. In this case, it is believed that CRISPR/Cas components enter predominantly into the nucleus [115]. It is assumed that this contributes to an increase in the level of editing, especially when a long donor DNA is added, for which the entry into the nucleus is difficult.

When working with RNP complexes and their electroporation into primary T cells, Nguyen et al. noted that the Cas9 protein in RNP is not very stable and can form aggregates [95]. Its stability is increased in the presence of an excess of gRNA, donor ssODN (short single-stranded DNA strands) [35,95], or polyanions, in particular poly-L-glutamic acid (PGA) [95]. The PGA molecule was used in three studies conducted by one research group [6,54,95], as well as several independent groups [49,55,102]. In the work by Nguyen et al., PGA led to a 4-fold increase in the level of knock-in for some loci in T cells and improved survival [95]. Kath et al. reported a 2- to 3-fold increase in knock-in levels, but did not observe a positive effect on cell survival [49]. Shy et al. reported only a positive effect on survival, whereas the effect on knock-in levels was weak [54]. Oh et al. did not observe any effect of PGA on either the level of knock-in or cell survival [55]. When T cells were edited with another nuclease, ErCas12a, in the presence of PGA, the level of CAR knock-in increased up to 3-fold and cell survival improved [102]. It thus can be concluded that, in most studies, PGA had a positive effect on the knock-in level for many loci.

Since both electroporation devices and commercial kits for use with them are expensive, alternative delivery routes are constantly being developed, one of which is the delivery of RNP using virus-like particles (VLP) [116,117,118]. Compared to electroporation, these particles may be less toxic to cells and would allow RNP to be delivered in vivo to specific target cells via pseudotyping. However, currently, VLP have only been used on primary T cells to introduce knockout [116]. Hamilton et al. achieved about 70% *TRAC* gene knockout in primary CD4+ T cells and 45% in primary CD8+ T cells. With electroporation, this level was 70% for both cell types [116]. Banskota et al. used VLP with a base editor and obtained 45–60% of cells with knockout of the genes *B2M* or *CIITA* [119]. For an efficient knock-in of a long construct, T cells could be transduced with a mixture of Cas-VLP and AAV6; this approach has been applied to CD34+ cells [120]. The possibility of efficient targeted packaging of donor DNA into VLP has not yet been demonstrated.

More recently, Foss et al. demonstrated an unusually high efficiency of RNP delivery into primary T cells using amphiphilic peptides based on HA2-TAT fusion, in particular the A5K peptide [100]. With peptide delivery, the *TRAC* gene knockout was 70% for Cas9 and 90% for Cas12a. In comparison, when the same RNP complexes were electroporated using the 4D Nucleofector device (Lonza, Basel, Switzerland), the knockout was 100% for both nucleases. To knock-in the αCD19 CAR T construct into the *TRAC* locus, the authors incubated the cells with the peptide and then added the AAV6 vector. The knock-in level reached 55% for RNP with Cas9 and 80% for RNP with Cas12a. The dsDNA donor was unsuitable for this purpose. The peptide had virtually no toxic effect on cells; as a result, the total yield of T cells with peptide delivery was two times higher than that with electroporation. The authors showed that the A5K peptide can be used for sequential editing, which can reduce the level of translocations that occur during multiplex editing [121]. When used with sequential electroporation, this task has no practical meaning because it causes the death of up to 80–100% of cells and is too expensive. Triple sequential knockout of *TRAC*, *B2M*, and *CD5* was 20% for peptide delivery and 30–40% for electroporation, but the overall cell yield was 14 times greater for peptide delivery [100].

4.Influence of NLS signals in the Cas nuclease on editing

It has been shown that Cas9 with an increased number of NLS is better delivered to the nucleus [62,122,123] and leads to increased editing levels [122,123]. Such data are available for cell lines and primary cells, as well as for the Cas9 and Cas12a nucleases [124,125]. In different studies, various combinations of NLS were used, most often combining SV40, nucleoplasmin, and c-Myc. A two-part NLS sequence produced a better result than a one-part sequence (from SV40) [124,126]. A common approach is to use a combination of one-part and two-part NLS signals and to include different NLS [123,124]. Some studies report that the location of NLS at the C-terminus of the protein is important for Cas12a [125,126], but this is not supported by other studies [127]. The effect of increasing the number or selection of a particular NLS may be stronger for Cas12a than for Cas9, as two NES motifs are predicted in the Cas12a sequence [126]. Thus, there are different interchangeable efficient constructs with a different set and location of NLS. It should be noted that all aforementioned optimizations of NLS in Cas nucleases have been performed in cell lines and have not been replicated specifically in primary T cells. Many studies on T cells use commercial nucleases Cas9 and Cas12a.The number of NLSs is unknown for these, but it is most likely two or more. When the Cas protein is delivered using amphiphilic peptides, more NLS signals can have an additional positive effect on the level of editing [100], possibly facilitating the entry of RNP into the cell [128].

#### 4.1.3. Donor DNA

The next component of the system that needs to be optimized is the donor construct, which serves as a template for homologous repair (see Figure 3). The donor can be used in different forms, such as ssDNA, dsDNA, plasmid DNA, and AAV. When making small changes to the genome, which can be up to several tens of nucleotides in length, short single-stranded DNA strands (ssODN) are more often used [35]. In these cases, the polarity of the chain and the asymmetry of the lengths of the homology arms can be important [129]. For large constructs, such as those encoding the CAR T receptor, all variants of donor molecules are used (see Table 2 and Appendix A).

The highest levels of knock-in, running up to 80%, are obtained by studies where donor DNA is delivered in the form of an AAV6 vector, even though these studies use various RNP electroporation devices. In these studies, which mostly used 4D Nucleofector for electroporation, knock-in with dsDNA reaches 45–50%. For plasmid DNA, approximately the same maximum level of knock-in is achieved. For ssDNA, a knock-in level of up to 10–20% is described. The levels of knock-in with dsDNA and ssDNA can be increased to up to 70–80% for some loci using tCTS modifications, which will be discussed below [54,95].

Donor DNA in the AAV6 vector

The AAV vector serotype 6 is most suitable for primary T cells [40,97,106]. AAV6 with donor DNA is added to T cells immediately after electroporation [101], within 10–15 min [56] or within 2–4 h [26,97]. In general, the advantages of this type of delivery are low toxicity as well as the ability to produce a vector and add it to the cells at a high dose [40,56]. Wiebking et al. demonstrated that the addition of AAV6 to T cells reduced survival by 11% [56]. If AAV6 was mixed with T cells after electroporation, the survival rate decreased by 27%, which was a good indicator compared to the results obtained in other studies (see Appendix A). When the authors tried to reduce the dose of AAV6, the knock-in reached a plateau at MOI 5000. The incubation time of cells with the virus appeared to be a more important factor: when it was extended from 30 min to 12 h, the level of knock-in increased 7-fold (toxicity was not estimated) [56]. Since all manipulations with the T cell genome are currently performed ex vivo, pre-existing immunity to AAV and the potential trigger of an immune response are not concerns. Unlike other forms of a donor molecule, AAV6 can theoretically be used for delivering a construct directly to T cells in vivo, and RNP complexes can be simultaneously introduced by virus-like particles [120].

The main disadvantage of AAV6 is the possibility of the non-targeted ligation of a construct at the DSB sites [40,96]. This was observed even in control cells without the induction of DSB via Cas9/gRNA [96]. The off-target incorporation of AAV can be significantly suppressed by using a Cas nickase, which has been demonstrated for T cells in the “spacer nick” approach [96]. In some studies, the non-target integration of AAV6 was not observed; however, the percentage of knock-in with errors at the ends of structures was high [26].

AAV production is a labor-intensive process that requires special equipment and reagents [132]. Authors who propose knock-in approaches based on the dsDNA donor note that these factors significantly limit the work and increase costs [49,54]. On the contrary, authors who use AAV6 for donor delivery [105] argue that the procedure for obtaining AAV has been established and that all parameters are optimized for GMP requirements [133].

2.Donor DNA in the linear form

In comparison to the AAV6 production, donor DNA in the form of dsDNA can be obtained more rapidly as a PCR product, a restriction plasmid product or, less commonly, as a circular plasmid DNA. To obtain a PCR product, highly processive polymerases are used, and the reaction product is additionally purified with magnetic particles [49]. With the development of molecular biology techniques, different methods have been proposed for achieving a highly efficient production of long ssDNA molecules. These methods have been successfully used to introduce knock-in into primary T cells [54], although this type of donor DNA is still more difficult to obtain compared to dsDNA.

Double-stranded linear donor molecules are significantly more toxic to cells than single-stranded molecules [27,54,129]. Compared to linear dsDNA, plasmid DNA can be more [27] or less toxic [55]. Toxicity can be substantially reduced via the co-administration of donor DNA with RNP into cells [27], but this evidence is not corroborated by other studies [13]. Oh et al. found that the use of nanoplasmids compared to conventional plasmids helped to reduce toxicity and produce a higher total cell yield [55].

In addition to the toxic effect on cells, donor DNA molecules can directly integrate into break sites. This risk is greatest for linear dsDNA molecules [27,99,129], which can form concatemers and be incorporated into the genome as multiple copies [99]. The modification of DNA ends was found to reverse this [134]; however, no effect has been observed on T cells following the addition of biotin or phosphorothioate to the 5′ ends of the donor DNA [13]. Plasmid DNA can also be inserted at DSB, presumably, when partially linearized [69]. For ssDNA, this risk is minimal [27,99].

3.Additional parameters of donor DNA

It has been noted that an increase in the dose of donor DNA leads to an increase in the level of knock-in, whereas the amount of Cas protein does not play a major role and can be reduced. However, in this case, toxicity increases sharply [49,54,130]. As a result, it is necessary to select the optimal dose to achieve the maximum knock-in level while maintaining cell viability.

For large donor constructs, with an increase in the length of homology arms, the knock-in level usually increases up to several times, but it gradually reaches a plateau at the length of 500–1000 bp. or more (depending on the design and varying in different studies) [13,55]. An increase in the length of homology arms and the length of the entire structure occurs simultaneously with an increase in toxicity and a decrease in cell yield [13,55]. For the delivery of an AAV6 donor, the vector’s capacity must be considered. For example, when inserting a construct with GFP, 1.3 kb homology arms gave a 30% higher knock-in than 0.6 kb homology arms, but at the same time filled the vector capacity to the limit [40].

The order in which CRISPR/Cas components are added can also be important. Roth et al. showed that the highest levels of knock-in can be achieved by first mixing RNP and donor DNA and then adding them to cells. However, this procedure causes maximum toxicity [27].

4.Time point for knock-in detection

Most studies on T cell editing used flow cytometry to assess the level of knock-in and knockout for the most common loci, whose manipulation is important for improving the properties of CAR T cells (see Appendix A). However, different studies used varying time intervals between delivering CRISPR/Cas components to cells and the assessment of the level of editing, ranging from 3 to 9 or more days. This complicates intra-study comparison, because with a short cultivation time, incomplete manifestation of knockout and knock-in may occur at the protein level. For example, in the study of Dai et al., the level of CAR T receptor knock-in in the *TRAC* locus was 40% on the third day of measurement and almost 80% on the ninth day [97]. A similar observation was made by Odé et al. [130].

5.Binding of donor DNA to Cas9 or gRNA and targeted delivery of donor DNA into the nucleus

Donor DNA must be present in the nucleus at the time of the cleavage of the target locus. The high efficiency of knock-in with donor constructs in the AAV6 vector is probably partially explained by the fact that the virus delivers the donor directly to the nucleus, where the capsid is unpacked and the DNA is released [135]. All other forms of donor constructs are incapable of directed transfer to the nucleus. In proliferating cells, the destruction and restoration of the nuclear membrane occurs cyclically, which permits the inclusion of the donor in the nucleus [136].

In order not to depend on this random process, it was proposed to use the RNP complex, namely the Cas protein, as a carrier of donor DNA into the nucleus because it contains the NLS [95]. Target sequences were introduced into the donor double-stranded DNA (CTS–Cas9 targeting sites), which the Cas9 protein binds due to gRNA complementarity. According to the authors, Cas9 serves as a shuttle and transfers the donor to the nucleus due to the presence of NLS. First, the authors used full-length irrelevant sequences of 20 nt in length and dCas9-NLS. The level of knock-in only increased 1.3 times when CTS were located on both sides of the donor in the PAM-in orientation. When 20 nt long CTSs with dCas9 RNP were used, the increase was up to 2-fold, but only for two CTSs in the PAM-in orientation. The authors then decided to use the same RNP complex to both edit the target locus and transport the donor into the nucleus. To prevent the cutting of the donor, truncated CTS (tCTS) sites of 16 nt in length were used. The authors referred to a review about the mechanism of DNA cleavage by Cas9 [63] and emphasized that such sequences are recognized but not cut by Cas9. However, the authors did not provide their own data on how such sites are cleaved in vitro or in cells. The addition of tCTS sites resulted in a 2–4-fold increase in the knock-in level of a 1.5 kb construct in different primary T cell loci.

Later, the same group of authors published a study, where ssDNA was used instead of dsDNA [54]. To create Cas9 binding sites, the authors added short oligonucleotides complementary to the target sequence, PAM, and a small fragment of the donor. The best result was obtained when the target sequence had 4–8 mismatches, which is consistent with the hypothesis that Cas9 recognizes but does not cut the donor. ssDNA+tCTS donors were tested on 22 different targets in primary T cells. In general, modified donors generated a better result, although with a high variability among loci, ranging from an almost complete lack of effect to a 10-fold increase in knock-in. The positive effect of tCTS sites was more pronounced for ssDNA than for dsDNA.

Smaller effects of tCTS modifications were obtained by two other research groups, who added two tCTS sites in the PAM-in orientation at the ends of a long donor [49,55]. Kath et al. used dsDNA donors with tCTS and noticed a slight increase in knock-in of no more than 20–30%. The overall level of editing remained the same, with a slight increase in the ratio of knock-in to knockout. The toxicity of dsDNA+tCTS donors significantly increased compared to unmodified donors with the same dose of DNA. As a result, the overall yield of edited CAR T cells did not change significantly. According to the authors, increased toxicity can be associated with the more successful delivery of DNA to the nucleus. Oh et al. found no effect from adding tCTS to dsDNA or nanoplasmids [55].

Other modifications to the donor construct have been described, leading to its spatial proximity with Cas9 or gRNA. For example, methods have been proposed to link donor ssODN molecules and gRNA with a covalent or non-covalent bond [137], or to cross-link donor DNA with Cas9 with a covalent bond (see the review [138]). However, to our knowledge, none of these approaches have yet been used in primary T cells.

### 4.2. Manipulating T Cell State

Genome editing efficiency and knock-in levels depend not only on the parameters of a genome editor and donor DNA, but also on T cell culturing conditions. Importantly, all manipulations should preserve the functionality and proliferative potential of T cells. Purification and activation methods should convert T cells into a state amenable to editing, which includes the efficient delivery and nuclear translocation of the CRISPR/Cas components, as well as rapid progression through the cell cycle required for HDR. Purification, activation, media composition, and cultivation time for primary T cells have been optimized thoroughly. Different small molecules have been tested to shift the balance between DSB repair pathways towards HDR. Recently, attention has been given to cellular response to Cas9, gRNA, and especially donor DNA, the suppression of which can potentially be beneficial for cell survival and the overall yield of edited cells. Key information on T cell state manipulations discussed in this section is summarized in Figure 4.

#### 4.2.1. T Cell Isolation and Activation

The main culturing conditions affecting T cell editing efficiency

T cell activation state is one of the key factors affecting T cell editing efficiency and, in particular, the level of knock-in. It was previously found that the genome editing of primary non-activated T cells is ineffective [90,139]. Accordingly, most studies on CAR T cell production and primary T cells editing now include the T cell activation step. There are several reasons for this. First, activated T cells survive longer in culture, which simplifies any manipulations with them [16]. Second, activated dividing cells undergo chromatin changes in many loci that are important for therapy, which improves the access of Cas9 to the editing site [140]. Third, successful HDR editing requires cells to undergo the S/G2 phase of the cell cycle [104]. Fourth, the activation and expansion of T cells in culture is necessary to obtain a sufficient number of cells to be administered to a patient [54]. Finally, cell activation is a necessary step of the protocol for the production of CAR-T cells via transduction because non-activated T cells are difficult to transduce [141]. Thus, on the one hand, to increase the level of editing, it is necessary to activate T cells. On the other hand, it is known that the functionality of CAR T cells in vivo is better when they are cultured in vitro for as little time as possible and have a less differentiated phenotype [142]. As a result, much effort is directed towards finding the optimal conditions for isolating and activating T cells, so that they can be successfully edited with the preservation of the phenotype of naive or memory cells [143]. Below are the main steps for isolating and activating T cells prior to editing. The next two sections provide information on editing non-activated T cells. References are given in the Table 3 and in a Appendix A.

First, T cells (total population, CD4+ or CD8+) are isolated from the donor peripheral blood mononuclear cells (PBMCs) using either positive or negative magnetic separation. The latter produces non-activated untouched cells. The original PBMCs are a much poorer starting material compared to T cells, producing fewer knock-in levels and lower cell yields [49]. The presence of monocytes in a preparation can slow down T cell proliferation [144]. In one study, a mixed population of cells was used for editing to obtain a cell product with modified αCD19 CAR T cells and, at the same time, preserve NK and γδT cells of the donor, which contributes to the control of recipient leukemia in haploidentical transplantation [56]. In this study, donor B cells were destroyed by CAR T cells during the expansion of the cell product.

Following isolation, T cells are most commonly activated with the αCD3/αCD28 antibodies that mimic activation signals on antigen-presenting cells. More physiological conditions are achieved when antibodies are immobilized on the surface [145]. Most often, magnetic beads with bound antibodies against CD3/CD28 are used, sometimes together with an antibody against CD2 (see Table 3). Antibodies adsorbed on plastic can be used instead of commercial beads [49]. With this activation method, the level of the CAR receptor knock-in did not differ from the level achieved with the activation with magnetic beads, and the cell yield was even higher. However, this study did not compare the functional characteristics of the cells activated using these two methods [49]. The possibility of standardizing and scaling up this method also remains unclear. In some studies, freshly isolated cells were first electroporated and then immediately activated [9]. However, no comparison was made with the standard protocol.

Various media and supplements can be successfully used for culturing T cells. Details can be found in a recent review [146]. In addition to stimulation via the T cell receptor and co-stimulation molecules, the presence of IL-2 in the medium is important for T cell activation. However, prolonged cultivation with IL-2 induces T cell differentiation and the acquisition of an effector phenotype [147]. The reduction in the IL-2 concentration or growth of cells with IL-2 for a short period of time helps to preserve a less differentiated phenotype [148]. The addition of soluble IL-7, IL-15, and IL-21 prevents differentiation into effectors and supports cell survival [149]. As a result, in most studies on T cell editing, a mixture of the cytokines IL-2, IL-7, and IL-15 is used.

When generating CAR T cells, it is very important that there are no tumor cells in the population of edited cells, since the transduction of a single tumor cell transduced with a CAR receptor can gain resistance due to antigen masking [150]. When producing cells resistant to HIV, it is necessary to culture cells in the presence of antiretroviral drugs to prevent the reactivation of the provirus [151].

In addition to the culturing conditions, another important parameter is the time from the moment of cell isolation to the moment of the introduction of CRISPR/Cas components. The most studied conditions for the delivery of CRISPR/Cas to T cells use RNP electroporation. In general, electroporation is carried out for 48 h after cell activation. In this case, both the maximum possible knock-in level and the maximum total yield of edited cells are achieved [55].

Experiments conducted with ZFN on cell cultures suggested that the level of editing can be increased by culturing cells at 30 °C or 32 °C for 24 h after the delivery of CRISPR components [152]. Later, this was observed for Cas9 on iPSC [153]. A temporary hypothermic step is included in some protocols to obtain edited T cells for clinical trials [8]. However, for primary T cells, we could not find direct comparisons of conventional culture conditions and conditions with temporary hypothermia. In addition, in one study carried out on cell lines, including the T cell line Jurkat, it was shown that the Cas9 protein, unlike ZFN and TALEN nucleases, causes higher levels of knockout and knock-in at 37–39 °C compared to 30–33 °C [154].

2.Knockout in non-activated T cells

It has been shown that cells survive in vivo better when they grow in culture for a short period of time and have a less differentiated phenotype [142]. Therefore, a few attempts have been made to edit freshly isolated naive T cells [82,83,84,155]. The introduction of knockout into primary non-activated T cells has been difficult to implement. Mandal et al. used plasmid electroporation and achieved only 5% knockout of the *B2M* gene. This level was increased to 18% with two sgRNAs per gene [139]. Next, Hendel et al. knocked out the *CCR5* gene in resting T cells [90]. When electroporated with plasmids encoding gRNA and Cas9, no editing was observed. During electroporation of Cas9 mRNA and chemically modified sgRNA, the knockout level varied from 6 to 20% [90].

More recently, studies that used RNP electroporation with a Lonza or Neon instrument demonstrated that naive T cells can be edited at various loci with knockout levels comparable to those of activated cells: from 20% to almost complete knockout [82,83,84,155]. This was facilitated by the observation that resting T cells were able to survive in culture for up to 6 weeks when cultured at a high density in the presence of IL-7+IL-15 [84].

A large-scale study was carried out by Seki et al., where the conditions for electroporation of resting T cells were selected by using the Lonza 4D nucleofector on the EH-100 program [83]. The efficiency of knockout of the *CXCR4*, *CCR7*, *CD127* and *IFNγ* genes in non-activated T cells ranged from 20 to 80%, depending on the gene and one of the three gRNAs used [83]. In a preprint by Aksoy et al., it was shown that non-activated T cells can be more successfully electroporated using a Neon device at more stringent parameters (2200 V 20 ms 1 pulse), in contrast to activated T cells (1600 V 10 ms 3 pulses). Under such conditions, an almost complete knockout of *CXCR4* and *CD127* occurred [155]. The same device and the same program (Neon, 2200 V 20 ms 1 pulse) were successfully used by Leoni et al. for *CD3* and *B2M* knockout [82]. Finally, in a recent study by Albanese et al. using the 4D nucleofector and the EH-100 program, the level of knockout of *PSGL-1* and *CD46* was about 90% with a single gRNA and about 99% with a pair of sgRNAs [84]. Furthermore, the authors achieved almost 100% efficiency when they knocked out 6 genes simultaneously, with two sgRNAs per gene. At the same time, the survival rate did not decrease significantly, and the cell phenotype for the main molecules remained unchanged.

Despite these successes, we identified only two studies that directly compared the level of editing between activated and non-activated T cells when applied to the same target genes [82,90]. In the study by Hendel et al., the knockout of *CCR5* in non-activated cells varied from 6 to 20%, whereas in activated cells it was about 50% with a much smaller variance between donors [90]. In the study by Leoni et al., the *CD3* and *B2M* knockout rates were 7% and 65% in non-activated cells and 60% and 30% in pre-activated cells, respectively [82]. This may be partially explained by a greater accessibility of the target locus to the nucleases, as previously suggested [140]. However, the authors underscored the importance of naming the turnover of a particular protein in the cell, which can be significantly accelerated during cell activation and will lead to a more rapid manifestation of knockout [82,83,84,156]. For instance, Albanese et al. noted that a complete *CXCR4* knockout occurred in one week, *CD46* knockout in two weeks, and *SAMHD1* knockout in more than four weeks [84]. Accurate assessment of the knockout level in non-activated cells at early time points requires an analysis at the genomic DNA level. At the protein level, the phenotype may appear after a week or more, which must be taken into account when performing functional tests.

3.Knock-in in non-activated T cells

HDR in mammalian cells occurs in the S/G2 phase of the cell cycle. It is thus generally accepted that editing via the HDR pathway cannot occur without active cell proliferation [15,104] and trying to edit non-activated T cells is not theoretically possible. However, two recent studies have contradicted these assumptions [13,84]. Albanese et al. edited freshly isolated T cells and achieved a 24% knock-in level in the locus of the first exon of the *SAMHD1* gene of the eGFP construct in the form of dsDNA [84]. At three other loci with a similar donor template, knock-in ranged from 1 to 10%. The authors also compared knock-in during conventional cultivation and cell activation after electroporation. If the cells were activated for 3 days after electroporation, the percentage of GFP positive cells increased 1.5–8 times, depending on the locus. According to the authors, this observation indicated endogenous regulation of GFP expression from this locus. It remains unclear what the mechanism of knock-in introduction into non-activated cells might be, and what kind of population among primary CD4+ T cells can be edited this way.

In the second study, the authors carried out knock-in in non-activated T cells of a construct with a promoter and a fluorescent protein at the *PD-1* locus [13]. According to flow cytometry data, the level of positive cells was 20% under activation conditions and about 10% under conditions without activation, although the control sample treated only with donor DNA without RNP was not shown in the latter case. The authors also compared three donors with different homology arms that direct the insertion of a construct via the HDR, MMEJ, or HITI pathways. Interestingly, for activated cells, the level of knock-in with the ‘HDR’ donor was 5 or more times higher than that with the ‘MMEJ’ or ‘HITI’ donors, and there was no difference for non-activated cells. This may indicate different mechanisms of construct incorporation in two types of cells. However, it is necessary to take into account the off-target insertion of a construct at blunt ends without HDR, which has not yet been analyzed [13].

Despite remaining questions, the editing of non-activated T cells is no longer impossible. Because knockout levels can be as high as those of activated cells, when cell activation is not desired, the editing of non-activated T cells can be performed. Knock-in into non-activated T cells has so far only been described in two studies [13,84], and further research is required to describe this observation in detail and explain its mechanism.

#### 4.2.2. Suppression of T Cell Response to Manipulation

DSBs produced by the Cas nuclease are recognized by the cellular repair system, triggering a DNA damage response. At the same time, components of the CRISPR machinery, namely, plasmid DNA, gRNAs, and donor DNA, are capable of activating the innate response to foreign nucleic acids. Depending on the type of cells and the strength of response, these changes can lead to either a delay or a complete block of the cell cycle and subsequent death, or a change in the functional state of cells, thus rendering the cells unsuitable for therapeutic purposes or even promoting tumor transformation.

Suppression of р53 activation

Genome editing with Cas9 activates p53 and then either causes cell death or significantly slows down cell division, including the division of T cells during the production of CAR T lymphocytes [53,157,158,159]. The greater the nonspecific activity of the gene editor, the greater the activation of p53 [159]. Apparently, this can occur in cells even without gRNAs in the presence of only the Cas9 protein [160]. When p53 was temporarily blocked with MDM2 co-transfection [157], dominant-negative р53 [158,159,161], or siRNA [162], the level of knock-in via HDR increased [161,163]. There are no such data for Cas12a.

The importance of p53 blockage has not been demonstrated for T cell editing. That said, one study, which was performed in a mouse model, supports this assumption [164]. The authors showed that the activation of p53 caused the death of edited non-activated CD8+ T memory cells. When p53 was inhibited by siRNAs, the cells remained viable.

Thus, the role of p53 blocking in the editing of primary T cells requires further study. This has clinical implications, since the activation of p53, as a result of editing with Cas9, reduces the yield of edited cells with the desired modification. At the same time, cells with genome damage and tumor suppressor mutations can be selected, which has been demonstrated in various cell lines [160,162].

2.Innate response to nucleic acids

CRISPR reagents are capable of activating nucleic acid receptors in T cells: direct sensors of nucleic acids induce degradation of DNA/RNA molecules, whereas innate immunity receptors activate synthesis of antiviral factors, inducing cell cycle arrest or even cell death [165]. For example, the gRNA molecule contains hairpins and 5′-triphosphate capable of activating RIG-like receptors [92], which was demonstrated in primary T cells [166]. When in vitro-transcribed gRNA was treated with phosphatase, cell viability improved, which may be critically important for T cell expansion ex vivo [167]. Modifications of nucleotides, which are easier introduced into chemically synthesized gRNAs than into IVT gRNAs, also suppress an innate response to gRNA [167]. The mode of gRNA delivery may also affect the magnitude of the innate immunity response. For instance, in 293T cells, a stronger induction of the *IFNB1* and *ISG15* genes was observed upon gRNA transfection with liposomes as compared to nucleofection [168]. Furthermore, the introduction of the Cas nuclease as mRNA activates an innate response in T cells [111]. Modifications to nucleotide bases, primarily uridine, prevent this process [169].

Finally, a repair template in the form of the ssDNA or dsDNA molecule is known to activate various DNA sensors [55,170] that can suppress T cell proliferation [171]. In the only study so far, the efficiency of knock-in upon CAR T cell generation was slightly increased after treatment with small-molecule inhibitors of DNA sensors, RU.521 and ODN A151 [49]. The combination of these inhibitors and the HDR enhancer Alt-R resulted in an additive effect, and the level of knock-in increased 1.5–2 times. However, the mechanism of action of these molecules remains unclear. It is possible that the effect of these substances is at least partially related to the effect on the cell cycle since more cells accumulate in the S phase after incubation with these DNA sensor inhibitors [49].

In summary, when considering innate immune response to nucleic acids, the delivery of CRISPR/Cas components in the form of the RNP complex of Cas nuclease and chemically modified gRNA is the safest approach. It is still not well understood how the suppression of the T cell response to nucleic acids contributes to the level of editing and the total cell yield.

#### 4.2.3. Manipulating NHEJ and HDR Pathways or Cell Cycle

To increase the level of knock-in by the HDR mechanism, it has been proposed to influence the choice of the DSB repair pathway, namely, to suppress NHEJ or stimulate HDR. For these purposes, attempts are being made to use various small molecules or fusion proteins of the Cas nuclease with factors that either block components of the NHEJ pathway or recruit members of the HDR pathway [138]. In addition, the cell cycle is manipulated to promote cell transition into the S phase, during which HDR is possible [172]. The published literature contains information on many such molecules and fusion constructs [138]; however, only several of these have been tested in primary T cells for the introduction of knock-in. Below, we summarize information from these studies [49,54,79].

The most pronounced effect was found for the M3814 and NU7441 molecules that block the DNA-dependent protein kinase (DNA-PK) and thus suppress NHEJ [54]. These molecules increased the level of knock-in 1.5-fold, either when using a short ssDNA construct, which allows to add an HA tag to the N-terminus of the CD5 protein, or when introducing a long ssDNA or dsDNA construct with the αBCMA CAR T gene into the *TRAC* locus [54]. An even stronger effect was found for the M3814 molecule in the study by Fu et al. [79]. In a knock-in model where a short insert was delivered as AAV6, the number of cells with the targeted replacement increased by a factor of 3. In a long knock-in model using AAV6 to deliver the mNeonGreen construct to the *EEF2* locus, the number of cells more than doubled. Based on the results of NGS, the authors showed that M3814 suppresses NHEJ by more than 90% and does not affect MMEJ [79].

In addition to the M3814 and NU7441 molecules, the NHEJ pathway is also blocked by commercial HDR enhancers, “HDR enhancer AltR v1 and v2”, as described by the manufacturer (Integrated DNA Technologies (IDT), Coralville, IA, USA). The HDR enhancers AltR v1 and v2 increased the level of knock-in of a dsDNA construct containing the CAR T gene in the *TRAC* locus from 25 to 35% and did not affect the appearance of off-targets [49]. The authors showed that these molecules should be added to warm media immediately after electroporation and that a three-hour incubation is sufficient for their maximum effect. In another study, the effect of the HDR enhancer in a short fragment knock-in model was more modest, standing at about 30% [54].

Another molecule tested on primary T cells is XL413, which inhibits the CDC7 kinase involved in cell cycle regulation [54,131]. In one study, XL413 increased the knock-in level of the GFP-encoding construct in-frame with endogenous loci up to two times [131]. In another study, XL413 increased the level of HA tag knock-in at the *CD5* locus 1.3–1.4 times [54]. Wienert et al. showed that blocking CDC7 using XL413 delayed cells in the S phase, which supposedly allowed them to increase the percentage of HDR, although a direct influence of this blockage on HDR proteins was not ruled out [131].

An even weaker effect was described for trichostatin A (TSA), an inhibitor of histone deacetylases [54,79]. In the model of a short knock-in of the HA tag at the *CD5* locus, TSA increased the number of HA-positive cells by only 15%. However, in combination with M3814, TSA resulted in a small additional increase in knock-in [54]. In another study, TSA did not affect the level of a short knock-in and increased the mNeonGreen knock-in to the *EEF2* locus by 1.3 times [79].

Since M3814, TSA, and XL413 have demonstrated effects when acting on different targets, Shy et al. decided to investigate their combinations [54]. To do so, a 1.5 kb ssDNA construct encoding tNGFR was knocked into 22 different loci either without small molecules or in the presence of mixtures of M3814+TSA or M3814+TSA+XL413. When the M3814+TSA or M3814+TSA+XL413 molecules were combined, most of the loci showed an increase in the knock-in level: whereas for some loci the increase was small, for others it was manifold. With a triple combination of the molecules, the knock-in level for all loci varied from 20 to 90%. However, during the production of CAR T cells on a large scale, the presence of these small molecules reduced the total cell yield by more than three times. The authors concluded that the use of these small molecules for clinical purposes may not be justified [54].

Indirect information on some other inhibitors that are frequently mentioned in the literature can be obtained from the data in a study by Kath et al., who tested a large panel of molecules on the Jurkat T cell line [49]. Here, the donor was a short ssODN designed to introduce the EcoRI site into the *HPRT1* locus. The result of editing was evaluated using NGS. A 2–3-fold increase in the knock-in level was found for the following molecules: Brefeldin A, Alt-R HDR Enh. V1, XL413, NU7441, Trichostatin A, CRISPY mix, Romidepsin, M3814, and Alt-R HDR Enh. V2. No effect was observed for the following molecules: Wortmannin, SCR7, L755507, and EPZ5676. The data on the Rucaparib and Pevonedistat molecules were not sufficient for an unambiguous conclusion. In addition, Fu et al. found no effect for the SCR7 molecule on primary T cells [79].

In summary, small molecules that block NHEJ, and in particular DNA-PK, allowed for the strongest increase in knock-in levels in primary T cells. At the level of specific indel sequences, it has also been shown that inhibitors, which block the formation of short indels and enhance indel patterns that are characteristic of MMEJ, increase the level of knock-in by the HDR mechanism [79]. However, the NHEJ mechanism is the first measure against DSBs, and, as a result, blocking NHEJ can be dangerous. Wen et al. showed that when NHEJ is inhibited, particularly in the presence of M3814, the number of large deletions caused by Cas9/gRNA increased in T cells. However, in the presence of a donor molecule, the number of large deletions decreased, which can supposedly outweigh the effect of NHEJ blocking [173].

## 5. Conclusions and Future Directions

In recent years, the field of primary T cell editing has expanded and achieved results that seemed impossible before. The knockout levels of genes, such as *TRAC*, *TRBC*, *B2M*, and *CIITA,* that are important for improving the properties of CAR T cells reach 90% or more. The knock-in levels of 2–3 kb constructs encoding the CAR T receptor are 80% in some studies. These results are now even possible to obtain in larger-scale experiments, as several studies have demonstrated the possibility of scaling up this technology to obtain hundreds of millions of edited T cells, which is the amount necessary for injection into a patient. These cells retain viability and functional properties both in vitro and in vivo. Clinical trials of T cells with knock-in are already underway (NCT04213469—[13,14], NCT03970382—[12], NCT05631912).

As the field of T cell genome editing via knock-in has entered clinical trials and more trials will be launched in the coming years, data on the safety and efficacy of the introduction of edited T cells in patients will soon be available. These data will provide some information on the level of knock-in required to achieve a therapeutic result that will guide future trials. Several advantages of knock-in as opposed to viral transduction have been suggested and experimentally validated in animal models, but its superiority has yet to be proven in clinical settings. Patient data will allow for the comparison of both T cell products from the patient’s perspective.

More attention should be given to off-target editing, imperfect knock-in with errors at the sides of the donor sequence, and their influence on T cell properties. High-fidelity nucleases [67], or possibly editing without DSB [174], will help to reduce the risk of unwanted edits.

Many different approaches have been suggested to increase knock-in levels, but not all of them have been applied in upscaling experiments. However, it is critically important to determine the utility of the promising approaches to large-scale manufacturing. For instance, large-scale experiments have indicated that small molecules, which affect DSB repair pathways or the cell cycle and are considered very promising, cannot currently boost the production of large numbers of modified T cells because these molecules significantly inhibit cell expansion.

The T cell response to the genome editing procedure, and especially knock-in, needs to be studied in more detail. This includes the role of p53 activation and the innate response to nucleic acids, as well as global changes in epigenome and metabolome. Results of such studies can help to better understand functional characteristics of the edited T cells and can potentially reveal molecular targets, the manipulation of which will increase knock-in levels.

Recent studies have shown that non-activated T cells can be edited via knock-in, which was previously considered impossible. Further research is needed to validate this and understand the mechanism behind this process. If proved, it may reveal new details about the DSB repair pathways.

Hopefully, these data will help to increase knock-in efficiency in human T cells and allow for the production of T cell products with improved functional characteristics for better therapeutical outcomes. 

## Figures and Tables

**Figure 1 biomedicines-12-00119-f001:**
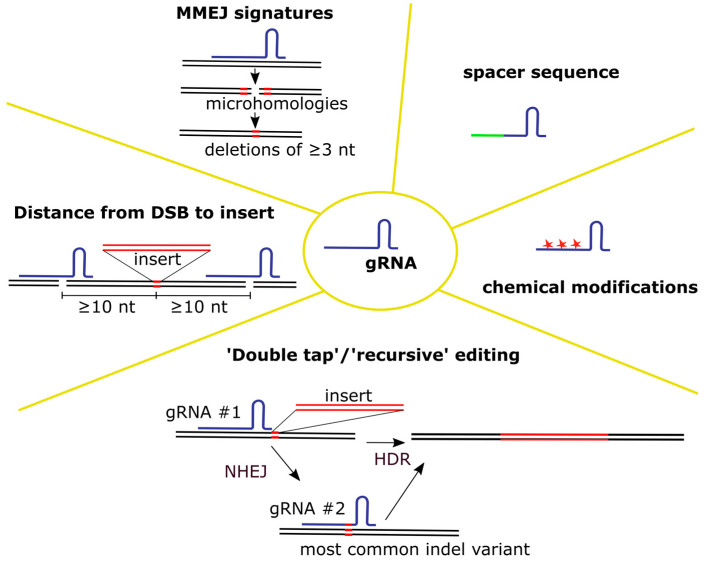
Selection of gRNA. Different parameters of gRNA affecting knock-in levels are depicted.

**Figure 2 biomedicines-12-00119-f002:**
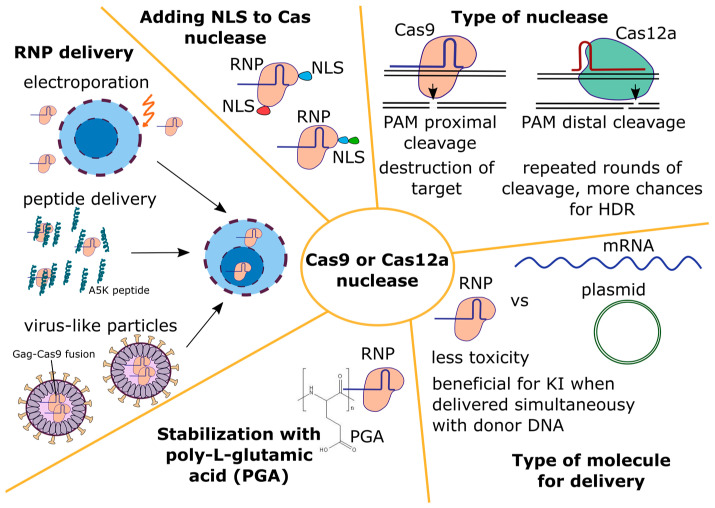
Different features of the Cas nucleases affecting editing efficiency.

**Figure 3 biomedicines-12-00119-f003:**
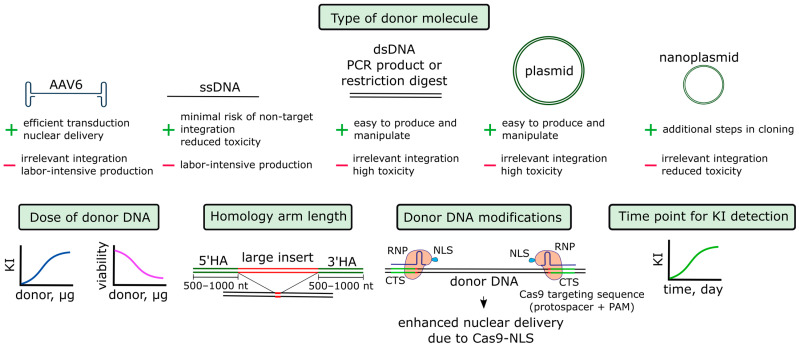
Different characteristics of donor DNA molecules affecting knock-in levels.

**Figure 4 biomedicines-12-00119-f004:**
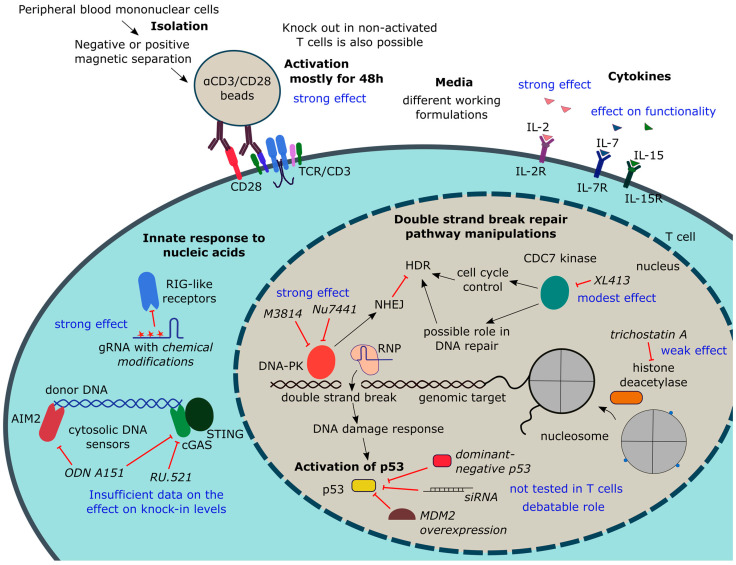
Manipulating T cell state for improving knock-in efficiency. The effects of certain stimuli on knock-in levels in primary T cells are depicted in blue.

**Table 1 biomedicines-12-00119-t001:** Factors affecting the editing efficiency of primary human T cells using CRISPR/Cas-mediated knock-in discussed in the manuscript.

Factor		Parameters
CRISPR components and method of their delivery into cells	gRNA	Spacer sequence
		Chemical modifications
		Using several gRNAs per gene
		Delivery method
		Protection from degradation
	Cas nuclease	Type of nuclease
		Type of delivered molecule
		Delivery method
		Nuclear localization signal (NLS)
	Donor DNA	Type of molecule
		Dose
		Length of homology arms
		Delivery method
		Modifications
Manipulating T cell state	Culturing conditions	Isolation
		Activation
		Media
		Activation time
		Cytokines
	T cell response to manipulation	Activation of p53
		Innate response to nucleic acids
	DSB repair pathway choice	Inhibition of NHEJ
		Stimulation of HDR

**Table 2 biomedicines-12-00119-t002:** Different types of donor DNA molecules used to edit primary T cells with CRISPR/Cas9-mediated knock-in of large constructs. Bold indicates two studies whose protocols were used to generate T cells for clinical trials.

The Type of a Donor DNA Molecule Delivered to T Cells	Reference
ssDNA	Roth et al., 2018 [27]
	Shy et al., 2023 [54]
dsDNA PCR product	Roth et al., 2018 [27]
	Nguyen et al., 2020 [95]
	Ode et al., 2020 [130]
	Kath et al., 2022 [49]
	Oh et al., 2022 [55]
	Shy et al., 2023 [54]
	Mueller et al., 2022 [52]
	Mohr et al., 2023 [102]
	Braun et al., 2023 [113]
	Glaser et al., 2023 [98]
dsDNA restriction digest	**Zhang et al., 2022** [13]
	Mohr et al., 2023 [102]
Plasmid	Wienert et al., 2020 [131]
	Oh et al., 2022 [55]
	**Foy et al., 2022** [12]
nanoplasmid	Oh et al., 2022 [55]
AAV6	Eyquem et al., 2017 [26]
	Dai et al., 2019 [97]
	Wiebking et al., 2020 [56]
	Fu et al., 2021 [79]
	Tran et al., 2022 [96]
	Foss et al., 2023 [100]
	Allen et al., 2023 [99]

**Table 3 biomedicines-12-00119-t003:** Conditions of T cell isolation, activation and culturing used in experiments on T cell editing with CRISPR/Cas. References are listed in a chronological order. Two papers whose protocols were used for T cell production in the clinic are highlighted in bold.

Parameter	Conditions	Reference
T cell isolation	No isolation from PBMC	Nahmad et al., 2022 [53]
		Kath et al., 2022 [49]
	Negative separation	Schuman et al., 2015 [35]
		Ren et al., 2017 [91]
		Eyquem et al., 2017 [26]
		Roth et al., 2018 [27]
		Webber et al., 2019 [85]
		Dai et al., 2019 [97]
		Nguyen et al., 2020 [95]
		Wienert et al., 2020 [131]
		Tran et al., 2022 [96]
		Mueller et al., 2022 [52]
		Balke-Want et al., 2023 [23]
		Shy et al., 2023 [54]
		Foss et al., 2023 [100]
		Mohr et al., 2023 [102]
	Positive separation	Ode et al., 2020 [130]
		Fu et al., 2021 [79]
		Kath et al., 2022 [49]
		Oh et al., 2022 [55]
		**Foy et al., 2022** [12]
		**Zhang et al., 2022** [13]
		Shy et al., 2023 [54]
		Glaser et al., 2023 [98]
Activation	Dynabeads Human T-Activator CD3/CD28 (Gibco, Thermo Fisher Scientific, Waltham, MA, USA) at a 1:1 or 2:1 bead:cell ratio	Ren et al., 2017 [91]
		Eyquem et al., 2017 [26]
		Roth et al., 2018 [27]
		Dai et al., 2019 [97]
		Webber et al., 2019 [85]
		Ode et al., 2020 [130] *
		Wienert et al., 2020 [131]
		Nguyen et al., 2020 [95]
		Wiebking et al., 2020 [56]
		Fu et al., 2021 [79]
		Zhang et al., 2021 [101]
		Kath et al., 2022 [49]
		Tran et al., 2022 [96]
		Shy et al., 2023 [54]
		Balke-Want et al., 2023 [23]
		Allen et al., 2023 [99]
		Foss et al., 2023 [100]
	αCD3/αCD28 soluble antibodies	Nahmad et al., 2022 [53]
	αCD3/αCD28-coated tissue culture plates	Schuman et al., 2015 [35]
		Kath et al., 2022 [49]
		Glaser et al., 2023 [98]
	T Cell TransAct (Miltenyi Biotec, Bergisch Gladbach, Germany)	Oh et al., 2022 [55]
		**Foy et al., 2022** [12]
		**Zhang et al., 2022** [13]
		Braun et al., 2023 [113]
	Human CD3/CD28/CD2 T cell Activator (STEMCELL Technologies, Vancouver, BC, Canada)	Mueller et al., 2022 [52]
		Mohr et al., 2023 [102]
Medium	OpTmizer CTS T cell Expansion SFM + 2.5% CTS Immune Cell SR (Gibco, Thermo Fisher Scientific, Waltham, MA, USA) + L-Glutamine + Penicillin/Streptomycin + N-Acetyl-L-cysteine (10 mM)	Webber et al., 2019 [85]
	RPMI 1640 + 10% FCS	Kath et al., 2022 [49]
		Glaser et al., 2023 [98]
		Braun et al., 2023 [113]
	RPMI 1640 + 10% FCS + 1× GlutaMAX (Gibco, Thermo Fisher Scientific, Waltham, MA, USA)	Tran et al., 2022 [96]
	RPMI-1640 + supplemented with 5 mmol/L Hepes + 2 mmol/L GlutaMAX (Gibco, Thermo Fisher Scientific, Waltham, MA, USA) + 50 μg/mL penicillin/streptomycin+ 50 μmol/L 2-mercaptoethanol + 5 mmol/L nonessential amino acids + 5 mmol/L sodium pyruvate + 10% FBS	Schuman et al., 2015 [35]
	X-VIVO 10 medium (Lonza, Basel, Switzerland) + 5% human serum + 1.6 mg mL^−1^ N-acetylcysteine + 2 mM GlutaMAX (Gibco, Thermo Fisher Scientific, Waltham, MA, USA)	Allen et al., 2023 [99]
	X-VIVO 15 medium (Lonza, Basel, Switzerland) + 5% FBS + 50 µM 2-mercaptoethanol+ 10 mM N-acetyl L-cysteine	Roth et al., 2018 [27]
		Wienert et al., 2020 [131]
		Nguyen et al., 2020 [95]
		Shy et al., 2023 [54]
		Foss et al., 2023 [100]
	X-VIVO 15 medium (Lonza, Basel, Switzerland)+ CTS Immune Cell Serum Replacement (Gibco, Thermo Fisher Scientific, Waltham, MA, USA)	**Zhang et al., 2022** [13]
	X-VIVO 15 medium (Lonza, Basel, Switzerland) + 5% human serum	Eyquem et al., 2017 [26]
		Dai et al., 2019 [97]
		Wiebking et al., 2020 [56]
	MEM-Alpha (Biological Industries, Sartorius, Göttingen, Germany) + 10% FCS	Nahmad et al., 2022 [53]
	For activation: PRIME-XV T Cell CDM media (Irvine Scientific, FUJIFILM Corporation, Santa Ana, CA, USA) For culturing: RPMI 1640 + 10% FBS + 2 mM GlutaMAX (Gibco, Thermo Fisher Scientific, Waltham, MA, USA) + 1 mM sodium pyruvate + 0.1 mM nonessential amino acids + 55 μM 2-mercaptoethanol + 100 U/mL penicillin + 100 μg/mLs streptomycin + 10 mM Hepes (Gibco, Thermo Fisher Scientific, Waltham, MA, USA)	Oh et al., 2022 [55]
	ImmunoCult™-XF T Cell Expansion Medium (STEMCELL Technologies, Vancouver, BC, Canada)	Fu et al., 2021 [79]
		Mueller et al., 2022 [52]
		Mohr et al., 2023 [102]
	TexMACS media (Miltenyi Biotec, Bergisch Gladbach, Germany) + 3% human male AB Serum	Balke-Want et al., 2023 [23]
		**Foy et al., 2022** [12]
	PRIME-XV (Irvine Scientific, FUJIFILM Corporation, Santa Ana, CA, USA)	**Foy et al., 2022** [12]
cytokines	IL-2	Schuman et al., 2015 [35]
		Eyquem et al., 2017 [26]
		Dai et al., 2019 [97]
		Wiebking et al., 2020 [56]
		Fu et al., 2021 [79]
		Nahmad et al., 2022 [53]
		Tran et al., 2022 [96]
		Mueller et al., 2022 [52]
	IL-7,15	Ode et al., 2020 [130]
		Oh et al., 2022 [55]
		**Foy et al., 2022** [12]
		Balke-Want et al., 2023 [23]
		Glaser et al., 2023 [98]
		Braun et al., 2023 [113]
	IL-2,7,15	Roth et al., 2018 [27]
		Webber et al., 2019 [85]
		Wienert et al., 2020 [131]
		Nguyen et al., 2020 [95]
		**Zhang et al., 2021** [101]
		Oh et al., 2022 [55]
		Kath et al., 2022 [49]
		Shy et al., 2023 [54]
		Zhang et al., 2022 [13]
		Allen et al., 2023 [99]
		Foss et al., 2023 [100]
		Mohr et al., 2023 [102]
Culturing time interval from isolation (day 0) until delivery of CRISPR/Cas (the next day after this interval)	1 d	Nahmad et al., 2022 [53]
		Balke-Want et al., 2023 [23]
		Allen et al., 2023 [99]
	2 d	Schuman et al., 2015 [35]
		Eyquem et al., 2017 [26]
		Ren et al., 2017 [91]
		Roth et al., 2018 [27]
		Webber et al., 2019 [85]
		Dai et al., 2019 [97]
		Wienert et al., 2020 [131]
		Ode et al., 2020 [130]
		Nguyen et al., 2020 [95]
		Wiebking et al., 2020 [56]
		Fu et al., 2021 [79]
		Kath et al., 2022 [49]
		Oh et al., 2022 [55]
		**Foy et al., 2022** [12]
		**Zhang et al., 2022** [13]
		Nahmad et al., 2022 [53]
		Mueller et al., 2022 [52]
		Shy et al., 2023 [54]
		Glaser et al., 2023 [98]
		Foss et al., 2023 [100]
		Braun et al., 2023 [113]
		Mohr et al., 2023 [102]
	3 d	Wiebking et al., 2020 [56]

* Activation and addition of cytokines 24 h after isolation.

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
