# Peer review of "Increasing Gene Editing Efficiency via CRISPR/Cas9- or Cas12a-Mediated Knock-In in Primary Human T Cells"

_biomedicines, 2024, doi:10.3390/biomedicines12010119_

Round 1
Reviewer 1 Report
Comments and Suggestions for Authors
The authors conducted a comprehensive review of genome editing in human T cells using the CRISPR/Cas9 or Cas12a system. Overall, the manuscript is well-structured, providing detailed information on the studies and appropriate discussions. However, several areas can be improved to enhance its quality.
1. The review's length could be condensed, particularly in section 4.2.1 covering T cell isolation and activation. For instance, paragraphs like lines 840-865 could be consolidated to improve conciseness.
2. Adding a section on future directions in addition to the conclusion would further enrich the manuscript.
Author Response
We thank all the reviewers for spending time and making valuable suggestions and commentaries which will significantly improve our manuscript. We hope that the revised version of the manuscript will meet the criteria for publication in Biomedicines journal. Please find below our point-by-point response to Reviewers’ commentaries.
Response to Reviewer 1.
Commentary #1
“1.The review's length could be condensed, particularly in section 4.2.1 covering T cell isolation and activation. For instance, paragraphs like lines 840-865 could be consolidated to improve conciseness.”
Response to commentary
Thank you for your suggestion. Indeed, the length of the section 4.2.1 is significant but we believe that this section is what mostly distinguishes our review from several previous reviews on this topic. In this section, we summarized key information on editing in non-activated T cells, especially knock-in in non-activated T cells, which was not thoroughly discussed in other reviews. Moreover, we analyzed the literature on the T cell response to manipulations including p53 activation and the innate response to nucleic acids. To the best of our knowledge, both areas have never been systematically discussed in the context of T cell genome editing. This allowed us to devote more space to not only mention the relevant studies but to highlight some details and provide some discussion which we believe will be of interest to the readers.
According to the suggestion by the Reviewer we edited the text in lines 835-871 to make it more concise.
Commentary #2
“2. Adding a section on future directions in addition to the conclusion would further enrich the manuscript.”
We agree with the reviewer that the manuscript needs a brief discussion of future research on this topic. Future directions are included in the ‘Conclusion’ section.
Reviewer 2 Report
Comments and Suggestions for Authors
1. This review summarizes and describes the parameters that affect human T cell editing efficiency using the CRISPR/Cas technology, with a special focus on gene knock-in. To streamline the site-specific editing of T cells with the CRISPR/Cas technology, several important parameters have been investigated and optimized, including components of the CRISPR/Cas machinery, namely the Cas nuclease, as well as the guide RNA molecule, the DNA donor template, and their delivery mode. Two main areas of T cell genome editing and acknowledging limitations of the clinically approved method of gene insertion using viral transduction. The review highlights features of two Cas nucleases most commonly used for T cell editing, which can potentially overcome the limitations of gene insertion using viral transduction. The review focusses on the DNA genome.
2. One area is how CRISPR/(d)Cas9 tools affect the epigenome in its engineering in the context of development and pluripotent stem cell self-renewal and differentiation, especially T cells and CAR T cells. The authors need to have a section on this important area of investigation.
3. For example, epigenetic states are initiated by environmental cues, developmental stimuli, cellular events and their sequelae and mediated by various cellular mechanisms including histone modifications, DNA methylation signals, non-coding RNAs and transcription factor networks. These mechanisms ultimately shape the chromatin architecture and fine-tune the transcriptome so that it can respond even in the absence of the initiating cue. These results in an epigenome that co-exists with the genome and is stably maintained in mitotic and post-mitotic cells, such as neurons and immune cells, in a self-sustaining fashion, which is a process that is critical for normal cellular physiology and function.
4. Clinical efficacy of CAR T-cell technology: limitations and challenges- 1) the T-cells used as starting material from patients are likely to have developed cancer associated T-cell dysfunction, which may not be reversible. 2) the procurement of normal donor leukocytes or iPSCs to produce a ‘universal’
CAR T-cell product that can be CRISPR-Cas9 engineered to overcome certain histocompatibility barriers and with enhanced persistence/antitumor function
will significantly improve the manufacturing of cellular immunotherapies and therapeutic durability. 3) the causes of unsuccessful CAR T-cell therapy are multifactorial and may not be addressed with synthetic biological improvements alone. Major limitations to successful CAR T-cell therapy occur
both during the manufacturing process in vitro and during CAR T-cell proliferation in vivo.
5. The review should also consider the following: 1) immunotherapy using CAR T cell therapies to target cancer stem cell biomarkers to solve this challenge. 2)
combining virotherapy and immune checkpoint inhibitors to induce an immunogenic TME to improve the outcome of immune checkpoint inhibitors in comn=bination with CAR T and CRISPR therapy. 3) the emerging approaches incorporating both genetic and epigenetic therapies combined with immunotherapeutic strategies need to be explored. 4) what are the challenges of this technology on the implications of the ongoing COVID pandemic and the challenges presented by COVID-19 for cancer
patients. 4) How do dietary and nutrigenomic approaches enhance and support the immune system, combined with immunotherapy, CRISPR therapy to
elicit a robust immune response against the tumor. 5) Considering the multi-mutagenic profiles of cancer, targeting metabolic syndrome could be considered an important therapeutic strategy for solid and non-solid tumors. The concept of metabolic syndrome is one of the critical survival mechanisms of cancer cells.
6. Challenges: Cancers have many compensatory pathways that can be engaged to overcome the therapeutic effects of conventional treatments and perhaps CRISPR therapy. Many of these compensatory pathways include activating survival pathways that can lead to an aggressive malignancy that is more challenging to treat during a followup round of treatment. As a result, a combination approach to target those pathways that account for cancer’s compensatory nature is mandatory. Perhaps, CRISPR/Cas9 screenings based on single detectable features (proliferation, expression of reporter proteins, membrane markers) is to isolate cell populations upon genetic disruption. However, recent efforts have been made to combine CRISPR/Cas9 with pooled gRNA libraries and single-cell RNA sequencing technology. These new approaches will help us better understand how genome regulatory regions affect the transcriptome at the single-cell level and perhaps how it affects epigenetic reprogramming.
Comments on the Quality of English Language
Moderate English editing is required.
Author Response
We thank all the reviewers for spending time and making valuable suggestions and commentaries which will significantly improve our manuscript. We hope that the revised version of the manuscript will meet the criteria for publication in Biomedicines journal. Please find below our point-by-point response to Reviewers’ commentaries.
Commentary #1
“2. One area is how CRISPR/(d)Cas9 tools affect the epigenome in its engineering in the context of development and pluripotent stem cell self-renewal and differentiation, especially T cells and CAR T cells. The authors need to have a section on this important area of investigation.
- For example, epigenetic states are initiated by environmental cues, developmental stimuli, cellular events and their sequelae and mediated by various cellular mechanisms including histone modifications, DNA methylation signals, non-coding RNAs and transcription factor networks. These mechanisms ultimately shape the chromatin architecture and fine-tune the transcriptome so that it can respond even in the absence of the initiating cue. These results in an epigenome that co-exists with the genome and is stably maintained in mitotic and post-mitotic cells, such as neurons and immune cells, in a self-sustaining fashion, which is a process that is critical for normal cellular physiology and function.”
Response to commentary
First of all, we would like to thank the Reviewer 2 for such thorough comments on the field of epigenome editing and CAR T cell technologies. We agree that the modification of the epigenome is a fascinating area of research and thanks to CRISPR/Cas9 system we have now unprecedented capabilities to manipulate chromatin state. Now it is possible to modify histone acetylation, methylation, phosphorylation with dCas9 fused to histone acetyl transferases (GCN5, TIP60 etc.), histone methyl transferases (PRDM9) or histone kinases (e.g. dCac9-dMSK1) and so on. Nevertheless, we believe that the topic of epigenome modifications with dCas9 lies outside the scope of our review. Our primary goal is to discuss methods and approaches to increase the efficiency of Cas9/12-mediated knock-in in T cells. Among aforementioned epigenome modifiers only PRDM9-Cas9 fusion was used to increase the efficiency of knock-in, but not in T-cells.
Commentary #2
“4. Clinical efficacy of CAR T-cell technology: limitations and challenges- 1) the T-cells used as starting material from patients are likely to have developed cancer associated T-cell dysfunction, which may not be reversible. 2) the procurement of normal donor leukocytes or iPSCs to produce a ‘universal’ CAR T-cell product that can be CRISPR-Cas9 engineered to overcome certain histocompatibility barriers and with enhanced persistence/antitumor function will significantly improve the manufacturing of cellular immunotherapies and therapeutic durability. 3) the causes of unsuccessful CAR T-cell therapy are multifactorial and may not be addressed with synthetic biological improvements alone. Major limitations to successful CAR T-cell therapy occur both during the manufacturing process in vitro and during CAR T-cell proliferation in vivo.
- The review should also consider the following: 1) immunotherapy using CAR T cell therapies to target cancer stem cell biomarkers to solve this challenge. 2) combining virotherapy and immune checkpoint inhibitors to induce an immunogenic TME to improve the outcome of immune checkpoint inhibitors in combination with CAR T and CRISPR therapy. 3) the emerging approaches incorporating both genetic and epigenetic therapies combined with immunotherapeutic strategies need to be explored. 4) what are the challenges of this technology on the implications of the ongoing COVID pandemic and the challenges presented by COVID-19 for cancer patients. 4) How do dietary and nutrigenomic approaches enhance and support the immune system, combined with immunotherapy, CRISPR therapy to elicit a robust immune response against the tumor. 5) Considering the multi-mutagenic profiles of cancer, targeting metabolic syndrome could be considered an important therapeutic strategy for solid and non-solid tumors. The concept of metabolic syndrome is one of the critical survival mechanisms of cancer cells.
- Challenges: Cancers have many compensatory pathways that can be engaged to overcome the therapeutic effects of conventional treatments and perhaps CRISPR therapy. Many of these compensatory pathways include activating survival pathways that can lead to an aggressive malignancy that is more challenging to treat during a followup round of treatment. As a result, a combination approach to target those pathways that account for cancer’s compensatory nature is mandatory. Perhaps, CRISPR/Cas9 screenings based on single detectable features (proliferation, expression of reporter proteins, membrane markers) is to isolate cell populations upon genetic disruption. However, recent efforts have been made to combine CRISPR/Cas9 with pooled gRNA libraries and single-cell RNA sequencing technology. These new approaches will help us better understand how genome regulatory regions affect the transcriptome at the single-cell level and perhaps how it affects epigenetic reprogramming.”
Response to commentary
Thank you again for your suggestions and outstanding expert opinion. We agree that all these areas of research have to be considered and thoroughly discussed in the context of CAR T cell manufacturing, usage and characterization. All these aspects can have a large impact on the quality of CAR T cells and the resultant therapeutic effect. However, we sincerely believe that all these topics lie beyond the scope of this review which focuses on the approaches to increase genome editing in T cells via knock-in and for the sake of clarity and structure we discuss only those parameters of genome editing that can boost the knock-in efficiency, therefore we omit post-editing events. Nevertheless, we have added a brief discussion of these issues in the section “Conclusion and Future Directions”.
Round 2
Reviewer 2 Report
Comments and Suggestions for Authors
1. The manuscript reveals over 666 grammar errors with 9% plagiarism which requires attention.
2. The authors appear to make changes in the revised manuscript but the changed text is the same. For example, in the introduction second paragraph, "In the former, T cells are edited to recognize and efficiently kill tumor cells [3]." This repeated again. The same type corrections are done throughout the manuscript. Please be precise.
3. The conclusions have not change from the origin.
4. The authors should consider adding a section that could affect the CRISPR technology as recommended by the reviewer. This is an important section to highly improve the manuscript and to make the field aware of the limitations and challenges. Here, a section on limitations and challenges would strengthen the manuscript.
Comments on the Quality of English LanguageThe manuscript reveals over 666 grammar errors with 9% plagiarism which requires attention.
Author Response
We thank the reviewer for spending time and making new suggestions on how to improve our manuscript. Please find below our point-by-point response to the comments.
Comments and Suggestions for Authors
- The manuscript reveals over 666 grammar errors with 9% plagiarism which requires attention.
We sincerely disagree with this comment. We checked the manuscript for grammatical errors with the Microsoft Word program which revealed zero errors. We also consulted with our colleague who speaks fluent English. We did not copy a single sentence from published literature. When it was necessary to repeat some information from other sources especially in the introductory parts, we always used our own wording and paraphrasing. We acknowledge that the manuscript can contain some stylistic errors, although we have tried to keep these to a minimum.
- The authors appear to make changes in the revised manuscript but the changed text is the same. For example, in the introduction second paragraph, "In the former, T cells are edited to recognize and efficiently kill tumor cells [3]." This repeated again. The same type corrections are done throughout the manuscript. Please be precise.
We made some changes only in the section 4.2.2. ‘Suppression of T cell response to manipulation’ part 2 ‘Innate response to nucleic acids’ lines 852-874 and in the section 5 ‘Conclusion’ which we changed to ‘Conclusion and future directions’ (lines 954-996). The rest of the document was left unchanged. The marks indicating some changes indeed appear throughout the text but they refer only to references. According to the reviewers’ suggestions we extended the Conclusion section and included 2 additional references in the conclusion. We use the Mendeley software to format references. After the addition of 2 new references the software updated the list and all the references and the text next to the references in the entire document became highlighted as changes, although they were not modified.
After the first round of the review, we uploaded the document with tracked changes. To simplify reading, this time we highlighted those changes and the added references in yellow.
- The conclusions have not change from the origin.
We modified the section 5. ‘Conclusion’ substantially after the first round of the review according to the comments of both reviewers and resubmitted the manuscript with tracked changes highlighted in red. In the second round of the review, we did not change this section but for clarity highlighted the changes introduced previously in yellow.
- The authors should consider adding a section that could affect the CRISPR technology as recommended by the reviewer. This is an important section to highly improve the manuscript and to make the field aware of the limitations and challenges. Here, a section on limitations and challenges would strengthen the manuscript.
Thank you for raising this important point. We briefly discussed the limitations of the CRISPR/Cas-mediated knock-in in the end of the section 3 ‘Genome editing with the CRISPR/Cas technology’ (lines 253-264). We also added a new reference to the study published on 21 November 2023 describing the influence of epigenetic state on the editing efficiency in primary T cells. We mentioned this study in the section 4.1.1. part 1 ‘Selection of gRNA’ (lines 305-305, 319-320). The text that we added in the second round of the review and the accompanying references are highlighted in blue.
We would like to note that some data on minimizing off-target editing and off-target donor DNA integration were discussed in the original version of the manuscript (lines 400-401, 540, figure 3). We also briefly mentioned that the chromatin state could affect editing efficiency (lines 299-300).